# Visceral Obesity in Non-Small Cell Lung Cancer

**DOI:** 10.3390/cancers14143450

**Published:** 2022-07-15

**Authors:** Lindsay Nitsche, Yeshwanth Vedire, Eric Kannisto, Xiaolong Wang, Robert J. Seager, Sarabjot Pabla, Santosh K. Patnaik, Sai Yendamuri

**Affiliations:** 1Jacobs School of Medicine and Biomedical Sciences, State University of New York, 955 Main Street, Buffalo, NY 14263, USA; lindsay.nitsche@roswellpark.org (L.N.); xwang242@buffalo.edu (X.W.); 2Department of Thoracic Surgery, Roswell Park Comprehensive Cancer Center, Elm and Carlton Streets, Buffalo, NY 14263, USA; yeshwanthreddy.vedire@roswellpark.org (Y.V.); eric.kannisto@roswellpark.org (E.K.); 3OmniSeq, 700 Ellicott Street, Buffalo, NY 14263, USA; robert.seager@omniseq.com (R.J.S.); sarabjot.pabla@omniseq.com (S.P.)

**Keywords:** adiposity, computerized tomography, epidemiology, image analysis, lung cancer, obesity, visceral fat

## Abstract

**Simple Summary:**

Body mass index may be an inappropriate metric for obesity in lung cancer as its use obscures the harm that obesity causes, which is noticeable when obesity is instead measured as visceral adiposity. Radiological assessment of visceral adiposity is practicable in lung cancer as patients undergo computerized tomography (CT) routinely, and this allows for the inclusion of visceral obesity as a meaningful variable in lung cancer studies. We used CT data to measure visceral adiposity of 994 non-small cell lung cancer (NSCLC) patients to obtain a profile of visceral obesity in NSCLC and associate it with demographic and tumor characteristics.

**Abstract:**

While obesity measured by body mass index (BMI) has been paradoxically associated with reduced risk and better outcome for lung cancer, recent studies suggest that the harm of obesity becomes apparent when measured as visceral adiposity. However, the prevalence of visceral obesity and its associations with demographic and tumor features are not established. We therefore conducted an observational study of visceral obesity in 994 non-small cell lung cancer (NSCLC) patients treated during 2008–2020 at our institution. Routine computerized tomography (CT) images of the patients, obtained within a year of tumor resection or biopsy, were used to measure cross-sectional abdominal fat areas. Important aspects of the measurement approach such as inter-observer variability and time stability were examined. Visceral obesity was semi-quantified as visceral fat index (VFI), the fraction of fat area that was visceral. VFI was found to be higher in males compared to females, and in former compared to current or never smokers. There was no association of VFI with tumor histology or stage. A gene expression-based measure of tumor immunogenicity was negatively associated with VFI but had no bearing with BMI. Visceral obesity is appraisable in routine CT and can be an important correlate in lung cancer studies.

## 1. Introduction

Obesity, the accumulation of excess or abnormal fat in the body that is harmful to its health, is most commonly gauged using the metric of body mass index (BMI). For a wide variety of diseases, increased risk and prevalence, pathophysiological changes, and worse outcomes are clearly linked with obesity as designated by a high BMI. Yet, a large body of evidence also shows a protective, beneficial effect of BMI-defined obesity for several diseases, such as chronic heart failure and chronic obstructive pulmonary disease [1]. This so-called obesity paradox is substantial for lung cancer, in which high BMI is a strong predictor of both lower risk and better prognosis, with meta-analyses suggesting hazard ratios (HR) of 1.12 per 5 BMI units [2] and 1.45–1.47 for BMI ≥ 30 [3,4], respectively. Obese lung cancer patients may also respond better to chemotherapy [5] and immunotherapy [6,7,8]. While the obesity paradox does not diminish the biomarker values of BMI, it confers a certain worth to the notion of being obese and goes against our understanding of the biological effects of harmful fat.

Resolving the paradox in lung cancer, to uphold the detrimental nature of obesity requires consideration of confounders, such as the negative effect of tobacco smoking on body weight and the greater use among the obese of commonly prescribed non-cancer drugs such as metformin and statins with anti-cancer effects [9]. But a major cause of the paradox appears to be the use of BMI to determine obesity [1,10]. BMI misclassifies obesity because it does not capture the heterogeneity of body fat, which is distributed among different types of adipose tissues (e.g., white and brown) and in different anatomic locations (e.g., subcutaneous and epicardial). About 80% of body fat in lean and healthy subjects is subcutaneous white adipose tissue, primarily in abdominal and gluteo-femoral regions [11]. It is biologically distinct [12,13] from intra-abdominal (visceral) white adipose tissue (5–20% of body fat). Subjects with more visceral fat relative to subcutaneous fat are at a greater risk for cardiometabolic disorders compared to subjects with a smaller visceral fat fraction [13,14]. Similar deleterious effects of visceral obesity likely exist for lung cancer. An examination of the NIH-AARP Diet and Health Study data showed that, unlike BMI, waist circumference (WC) was positively associated with higher lung-cancer-specific mortality (HR = 1.8 for highest vs. lowest WC categories) [15]. Increased risk of lung cancer has also been associated with larger WC, with relative risk or HR of 1.1 per 10 cm of WC in pooled analyses of 0.8–1.6 million subjects [2,16]. The WC-based A Body Shape Index (ABSI) metric, which also uses age, sex, weight, and height for its derivation, has been associated with a higher risk of developing lung cancer (HR = 1.2–1.4) [17,18].

As a measure of visceral adiposity, WC and some other anthropomorphic metrics of obesity, such as waist-hip ratio, may be better than BMI but less precise than imaging methods such as dual-energy X-ray absorptiometry (DEXA) and computerized tomography (CT) [19,20]. Visceral obesity identified by DEXA is associated with increased lung cancer risk, with HR of 2.0 for highest vs. lowest quartiles [21]. CT-based assessment of visceral obesity is very practicable for lung cancer patients as they routinely undergo CT, and user-friendly software to quantify visceral obesity from CT scans is available [22]. Visceral obesity measured using CT was associated with poor prognosis of lung cancer treated with chemotherapy in a study of 200 patients [23]. Recent work by our group, examining visceral obesity in CT scans of 513 patients, has shown similarly poor prognosis for surgically treated stage I/II non-small cell lung cancer (NSCLC) patients with high visceral obesity [24]. Here, to characterize for the first time the prevalence of visceral obesity in NSCLC and its associations with demographic and tumor features using a large cohort, we present a profile of visceral obesity that was obtained from 994 NSCLC patients using their CT scans. We hope that our work will encourage and facilitate the inclusion in clinical and translational lung cancer studies of visceral obesity metrics that are readily available to surgeons and physicians in routine CT data.

## 2. Materials and Methods

### 2.1. Clinical Data of the NSCLC Cohort

Institutional thoracic surgery databases and cancer registries were searched to retrospectively identify an arbitrary set of about a thousand patients with a diagnosis of primary NSCLC who underwent treatment at Roswell Park Comprehensive Cancer Center in Buffalo, New York, USA during 2008–2020. In all, 994 such patients were identified. Information on age, sex, race (African-American, Asian, Caucasian, or other), history of tobacco smoking (current, former, or never), body mass index (BMI), percent-predicted values of forced expiratory volume at 1 second (FEV_1_) and diffusion capacity of lungs for carbon monoxide (DLCO) of patients, and pathological stage as per the 7th or 8th editions of the staging manual of the American Joint Committee on Cancer and Histology (adenocarcinoma, squamous cell carcinoma, or other) of NSCLC tumors at the time of diagnosis was collected from patient medical records. We could not use the assigned stage with the same edition of the staging manual because of data unavailability.

### 2.2. Assessment of CT Images for Visceral Obesity

This work was performed by authors E.K. and X.W. CT scans that were obtained for pre-operative workup for tumor resection or closest to the biopsy date were analyzed. A single axial cross-sectional image at the level of third lumbar vertebra (L3) was identified in the scan data. In case a scan did not extend to the L3 level, images at the L2 or L1 level were identified. Only images that covered the entire body section and that were subjectively deemed to be of high quality were utilized. To quantify adiposity in the image data, ImageJ software (version 1.52 Java 1.8.0_112, National Institutes of Health, Bethesda, MD, USA) with Bio-Formats plugin (version 6.4.0) was used as per a method that we have applied in the past [24]. The entire cross-section of the patient was selected with either the wand or polygon tool of ImageJ. Total fat area (TFA) within this region of interest (ROI) was identified through subjective manual thresholding of Hounsfield unit (HU) values of the image pixels, typically the −150 to −50 HU range, to select fat but not muscle or other tissue. If necessary, the polygon tool was used to selectively deselect areas that were considered to be within the intestinal lumen. Subcutaneous fat area (SFA) was similarly measured but with the abdominal wall as ROI (see Graphical Abstract). Visceral fat area (VFA) was calculated as the difference between TFA and SFA. Visceral obesity was estimated as the ratio of VFA to TFA (visceral fat index, VFI). Individual values of these measures are provided in Appendix A. Accuracy of the ImageJ-based fat area assessment method was examined for a set of 54 patients whose images were assessed by author Y.V. using sliceOmatic software (version 5.0 Rev-16c; Tomovision Software, Magog, Canada) with ABACS+ plugin (Rev-1.0.0; Voronoi Health Analytics, Vancouver, Canada). Briefly, images were labeled with their vertebral level prior to auto-segmentation of image pixels by the plugin. Manual corrections to the segmentation were made in the plugin’s *Region Growing* mode as necessary. The sum of *Visceral Adipose Tissue* and *Intra-Muscular Adipose Tissue* areas determined by the software was considered as VFA.

### 2.3. Targeted Gene Expression Profiling of NSCLC Tumors

For a subset of the cohort’s subjects, tumor transcriptome data had been obtained to guide therapy. This gene expression profiling was performed by OmniSeq (Buffalo, NY, USA) using US Clinical Laboratory Improvement Amendments (CLIA)-licensed platforms and as per preexisting operating procedures (i.e., the work was not conducted for the purpose of this study). Briefly, hematoxylin-eosin-stained sections of formalin-fixed paraffin-embedded (FFPE) tumor tissue blocks were evaluated by a pathologist to ensure ≥5% tumor content and ≤50% necrosis. RNA was extracted from the FFPE samples for targeted sequencing of 397 cancer- and immunity-related genes [25]. Gene expression was derived from the sequencing data after background subtraction, normalization against a set of housekeeping genes, and conversion to percentile ranking with respect to a reference set of 735 solid tumors of 35 histologies [25].

### 2.4. Statistical Analyses

Visualization of tumor gene expression was performed using unsupervised hierarchical clustering with Pearson’s correlation as the distance metric to identify patient clusters, followed by unsupervised k-means clustering with k = 3 to group genes into three stable clusters that largely consisted of either 17 cancer/testis antigen (CTA) genes, 161 genes associated with inflammation, or 184 other immune and cancer genes. A tumor immunogenic signature (TIGS) was calculated for each patient by averaging the expression of the 161 inflammation-related genes, and the scores categorized as strong (> 61), moderate (43–61), and weak (<44), based on our prior observations obtained using unsupervised gene expression analysis of 1323 solid tumors spanning 35 histologies [26]. All analyses were performed using R (versions 3.6 and higher). For two-group comparisons, Welch t or Wilcoxon rank sum and Fisher’s exact tests were respectively used for continuous and categorical variables. Welch ANOVA test with the Games–Howell procedure was used in comparisons involving more than two groups. For multivariate analysis, generalized linear modeling with maximum likelihood estimation was used. R and Prism (version 9.3.1 for macOS; GraphPad Software, San Diego, CA, USA) were used for graphing. Unless noted otherwise, default values were used for statistical software options, tests were two-tailed, and the threshold of 0.05 was used to deem significance from *p* values. This study is reported as per Strengthening the Reporting of Observational Studies in Epidemiology (STROBE) guidelines for cohort studies (Appendix A).

## 3. Results

### 3.1. Measurement of Visceral Obesity with Fat Tissue Areas of Abdominal CT Images

We retrospectively quantified visceral obesity in 994 NSCLC patients by measuring abdominal fat tissue areas in axial cross-sectional CT images (Appendix A). There were 559 (56.2%) females among these patients, most of whom (89.2%) were Caucasian (white), with 6.9% African-American (black) and 1.3% Asian. NSCLC histology was adenocarcinoma in 61.8% and squamous cell carcinoma in 26.3% of the patients. The tumors were at pathological stage I, II, III, and IV in 55.2%, 24.3%, 11.7%, and 8.7% of the cases, respectively. The CTs that were used for fat measurement had been obtained within a year of tumor resection or biopsy (mean = 0.1 year). Fat tissue was quantified in the entire abdominal section (TFA) or in its subcutaneous region (SFA). The difference (TFA – SFA) was regarded as visceral fat area (VFA), and visceral fat index (VFI) was defined as the VFA/TFA ratio. These measurements were obtained at one or more of L1, L2, and L3 vertebral levels. CT data at the L3 level were unavailable for 4.6% of the patients.

We had assessed the accuracy of our ImageJ-based fat area measurement method in the early phase of our work by also measuring fat areas in 54 CT images with another method that uses the sliceOmatic software [27]. Examination of linear regression slope, Pearson correlation coefficient, and Bland–Altman bias showed that the two methods had good agreement for all four fat measures (SFA, VFA, TFA, and VFI), with slope, correlation coefficient, and bias values of 0.96–1.02, 0.98–0.99, and 5.4–15.3%, respectively (Figure 1). The ImageJ method was also judged for observational error in the early phase of our work by evaluating the concordance of measurements of 53 CT images made independently by two observers. Bland–Altman bias values in this analysis were 1.8%, 6.1%, 3.3%, and 3.1%, respectively, for SFA, VFA, TFA, and VFI (Figure 1).

### 3.2. Visceral Obesity at Different Body Levels

We also studied variation in fat area across L1–L3 vertebral levels in the early phase of our work by examining 46 arbitrarily chosen cases. Compared to L3, VFA was smaller at L1 and L2 levels by an average of 7.4% (standard error [SE] = 2.5) and 0.7% (1.9), respectively (Figure 1). The average reduction in SFA was greater, 29.6% for L1 (SE = 1.7) and 14.0% for L2 (1.7). The net effect of these differences was a larger VFI at L1 (by 13.8% on average; SE = 2.2) and L2 (7.4%; 1.5) compared to L3. The variation in VFI across the vertebral levels was significant enough to suggest that an L1 or L2 VFI value should not be used in lieu of an L3 value when CT data do not cover L3, as may be the case with many chest CT scans. In our cohort, use of L1 or L2 instead of L3 VFI, respectively, caused misassignment of 34.7% and 13.0% of cases to top instead of bottom L3 VFI-based tertile and vice versa, respectively.

### 3.3. Change in Visceral Obesity over Time

The relative stability of BMI has been well-documented in previous studies, e.g., [28]. We sought to assess if the same held true for visceral obesity. For 37 patients, we were able to additionally measure abdominal fat areas in CT images from two to five different time-points over a period of three years. The areas were measured at L1, as most of the CT scans did not include the L3 level. Analysis of the time series data, which did not consider clinical events such as surgery and cancer recurrence, showed that among the four types of fat measures, VFI was the most stable over time, with 61% of values staying within 10% of the initial VFI during the three-year period (Figure 2). For SFA, VFA, and TFA, 40%, 33%, and 28% of values were within the ±10% range, respectively. BMI at multiple time-points was unavailable for these patients, so time stability of BMI was assessed in a separate subgroup of 70 patients for whom serial BMI data were available (two to 35 additional time-points within three years). Among these patients, 74% of the BMI values stayed within the ±10% range (Figure 2).

### 3.4. Associations of Visceral Obesity with Demographic and Clinical Features

Considering the relative stability of abdominal VFI over time, compared to SFA, VFA, or TFA, we used this metric to correlate visceral obesity with different NSCLC patient and tumor characteristics. VFI at the L3 level was used as it was available for the majority of cohort (n = 947). Features of this cohort are summarized in Table 1. Median and inter-quartile range (IQR) of VFI was 0.46 and 0.35–0.58. VFI was significantly higher among males (median [IQR] = 0.58 [0.50–0.64]) compared to females (0.38 [0.30–0.45]) with Welch t test *p* value < 0.001 (Figure 3 and Figure 4).

This large difference for VFI arose from smaller SFA (26% for median value) but larger VFA (79%) of males compared to females (Appendix A). The male and female subgroups did not differ significantly for BMI (*p* = 0.05) but did so for age (*p* = 0.003) and smoking history (*p* = 0.001). The VFI difference between males and females remained significant when adjusted for these two factors in multivariable regression analysis (β coefficient for male sex = 0.18; SE = 0.01). In both males and females, BMI had minimal correlation with VFI, with respective Pearson correlation coefficients of 0.12 and −0.05 (Figure 3). The VFI-BMI correlation coefficient was modestly higher for the former smoker subgroup of males (0.17) but not females (−0.04). Correlation of VFI with age was positive for both sexes but modest (0.21 and 0.32, respectively; Figure 3).

Both VFI and BMI were similar among subjects of white, black, and Asian races (Welch ANOVA *p* values > 0.05; Figure 4). Current and never smokers had similar but significantly smaller VFI than former smokers (*p* = 0.005 and 0.006, respectively). On the other hand, BMI of current smokers was smaller than that of both former (*p* < 0.001) and never (*p* = 0.025) smokers (Figure 4). Neither VFI nor BMI had any association with pathologic stage of tumor (*p* > 0.05). BMI had no association with tumor histology, whereas patients with squamous cell carcinoma histology had a higher VFI compared to adenocarcinoma (*p* < 0.001; Figure 4). When adjusted for sex and/or smoking history, there was no association of VFI with histology (Wald *p* > 0.05 in multiple logistic regression). In generalized linear modeling for multivariable analysis of the associations with VFI, age, sex, and race but not smoking history were found to be independently associated with VFI, with β coefficients of 0.003 (Wald *p* = 9.7 × 10^−14^), 0.18 (male vs. female; *p* < 10^−16^), and −0.07 (black vs. white race; *p* = 5.6 × 10^−6^), respectively. In contrast, smoking history and race but not age or sex were associated with BMI in similar analysis, with β coefficients of 2.81 (former vs. current smoker; *p* = 6.1 × 10^−9^), 2.60 (never vs. current smoker; *p* = 9.6 × 10^−4^), and −5.31 (Asian vs. white race; *p* = 7.9 × 10^−3^). Two lung function measures that were examined, DLCO and FEV_1_, had no association with either VFI or BMI in linear regression models that included sex and age as covariates (all *p* > 0.05).

### 3.5. Variation of Tumor Inflammatory Gene Expression with VFI and BMI

We report this observation to illustrate that biomarker potential and biological insight can emerge when obesity is judged by visceral adiposity and not BMI. A subset of patients of our cohort had their tumors profiled for the expression of 397 cancer- and immunity-related genes for the purpose of clinical care guidance. We evaluated this gene expression data to explore the association of visceral adiposity and BMI with tumor biology. Because our previous work has shown that the use of metformin or statin affects NSCLC tumor biology [9], we excluded patients who had been prescribed these drugs. The 175 patients, all stage III/IV treatment-naive cases, whose data were thus analyzed, were grouped by their L3-level VFI into tertiles using cutoff values determined with the full cohort (0.39 and 0.53). Clustering of gene expression data showed separation of cancer/testis antigen genes from a set of 161 inflammation-related genes whose joint expression is known to be associated with tumor immunogenicity and response to immune checkpoint inhibitors for multiple cancers [26] (Figure 5A). The expression of these 161 genes is summarizable as tumor immunogenic signature score (TIGS) and reflects activities of B and T cell activation, interferon γ, and other cytokine pathways. In NSCLC, patients with high TIGS score have improved survival and responsiveness to immunotherapy [26]. In our cohort, patients with greater visceral obesity (VFI in top tertile) had a significantly lower TIGS score compared to patients with VFI in the bottom or middle tertiles, with Wilcoxon rank sum test *p* values of 0.04 and 0.02, respectively (Figure 5B). TIGS scores were not significantly different between patients categorized by BMI (*p* = 0.18 for overweight/obese [BMI ≥ 25] vs. others; Figure 5B).

## 4. Discussion

In this study, we profiled visceral obesity in NSCLC by examining about a thousand patients with the disease. While very few studies [23,24,29], all involving a few hundred subjects, have looked at visceral obesity in lung cancer, they show that visceral adiposity, unlike BMI [1,3,10], truly portrays the harm of obesity in lung cancer and is the more appropriate variable to include in epidemiological and clinical studies. The negative association of tumor immunogenicity with VFI but not BMI that we observed (Figure 5) also illustrates this point. Besides surveying CT-based visceral fat measurements and associating them with some important demographic and pathological variables in NSCLC, our study demonstrates the practicability and encourages the use of routine CT data to include visceral obesity as an examined factor in lung cancer research. Visceral adiposity is also of relevance in immediate clinical care of lung cancer for its influence on pulmonary function [30] and outcomes such as post-pneumonectomy complications [31] and radiation pneumonitis [32].

Reference values for SFA, VFA, and VFI for different populations, either healthy or with a disease such as lung cancer, have yet to be established. A step towards this direction is a recently published CT-based survey of SFA and VFA of about 1700 healthy adults in the USA [33]. The reported average SFA and VFA values are lower than those of our cohort (Welch t *p* < 0.05 for both sexes), as might be expected from the different age and health status of the two cohorts. In both and many other studies, intra-muscular fat was considered visceral fat (VFA), which was defined as non-subcutaneous abdominal fat and derived as the difference of TFA and SFA. Examination of 38 arbitrarily selected cases of our cohort showed that, at the L3 level, intra-muscular fat constituted an average of 18% of VFA (SD = 14%). The large difference between the sexes for fat areas and VFI (Figure 2 and Figure 3) observed in our study is well-known. Separating the effect of VFI from that of sex on a response of interest may therefore require subgroup or interaction analysis. We did not observe any association between VFI and tumor histology or stage, but, like BMI, VFI was negatively affected by current tobacco smoking (Figure 4).

We could not evaluate how well WC and waist-hip ratio reflected visceral adiposity of the study’s subjects since information for these anthropomorphic measures was unavailable to us. Literature suggests that the two measures are inadequate proxies of visceral obesity, though they are better than BMI, e.g., [34]. Compared to other methods, including DEXA and magnetic resonance imaging, CT has the best specificity, accuracy, and reproducibility for measuring visceral adiposity [35]. Although volumetry of abdominal fat with CT is ideal for assessing visceral obesity and is relatively easy with existing software, it requires full CT coverage of the abdomen and the larger amount of data takes more time to process. We used axial cross-sectional CT slices to semi-quantify visceral obesity from abdominal fat area measurements. We summarized the primary CT measurements of SFA and VFA into VFI, the visceral fraction of abdominal adiposity, as the visceral obesity marker. The VFI metric has been used in multiple studies and has been shown to be biologically and clinically meaningful. However, other SFA- and VFA-based visceral obesity indices, such as VFA per third power of height, may be stronger correlates of diseases or have smaller sex bias than VFI [36]. We had incomplete height and weight information for our cohort and could not explore indices other than VFI.

SFA and VFA of the abdomen vary along the body axis (Figure 1). Studies indicate that VFA at the L3 vertebral level correlates the best with visceral fat volume, with Pearson r > 0.95 in Caucasians as well as African-Americans of both sexes [37]. At L1, the correlation coefficient is slightly lower, though >0.90, and varies more among the different races and sexes. Correlation of SFA with total abdominal subcutaneous fat volume is high, with r > 0.95, for both races and sexes at both L1 and L3 levels [37]. Thus, while VFI measurement may be ideal at L3, an L1-level measurement is a reasonable metric for visceral obesity if a CT scan does not reach L3, as may be the case for routine chest CT. VFI at L1 and L3 levels, however, is not equivalent (Figure 1), and the body level at which VFI is obtained has to be considered in data analysis. It should be noted that slice thickness and use of contrast agent can influence fat measurements of CT images to a minor degree, as does the subject’s respiratory phase, especially at the L1 (but not L3) vertebral level [33]. As such, the cross-sectional abdominal visceral fat area is only a correlate and not an accurate and precise portrayal of abdominal visceral fat volume. A small study of lung cancer patients observed that the coefficient of determination (r^2^) for the two types of measurements was 0.75 [38].

An important limitation of our visceral obesity profiling is that the NSCLC cases were predominantly Caucasian (89%) and were from northeastern USA. The years of diagnosis spanned 12 years and, considering the increasing prevalence of obesity worldwide over time, the distribution of visceral obesity among NSCLC patients that we observed is likely different from that of an earlier time period. Another issue with our study is that for many of the patients, the CT image that was used for fat measurement had been obtained up to a year before or after surgery. This was because of unavailability of a CT scan from a time closer to surgery with adequate coverage of the abdomen. Our limited survey of VFI stability over time (Figure 2) shows that fluctuation in VFI values of ±10% over the period of a year is possible. How lung cancer or its treatment affects visceral obesity, including indirectly, such as through smoking cessation following cancer diagnosis, remains to be examined. Late-stage NSCLC may be associated with a reduction in total body fat [39]. Examination of the effect of VFI on disease outcome was not an objective of our study. In addition, the duration of post-surgery follow-up was less than three years for about a quarter of the study cohort, and we have previously reported outcome analysis for a majority of the cohort’s stage I/II cases, showing worse overall and recurrence-free survival of patients with high VFI [24].

The goals of our study were to draw a picture of visceral obesity in lung cancer and highlight its associations with common demographic and clinical features to aid analytical considerations in future lung cancer studies. We also wanted to point out the practicability of measuring visceral obesity by examining routinely collected radiological data using facile software-based methods. CT image data and workflow that are used for visceral obesity examination can be simultaneously used to measure other body composition features, such as skeletal muscle area [31], which may also have biomarker value in clinical care and research. With advances in image analysis, including incorporation of deep-learning-based approaches [40], it is possible that in the very near future, visceral obesity metrics will be automatically and accurately generated within radiology viewer applications themselves. Measurement of visceral obesity and associating these measurements with lung cancer outcomes in very large cohorts will be necessary to identify clinically meaningful cut-offs for CT-based fat measurements, as has been the case for BMI for many diseases. These studies may also identify equivalent cut-offs for more accessible markers of visceral obesity such as ABSI, WC, and the serum-biochemistry-based visceral adiposity index (VAI) [41]. Automated approaches to quantify visceral fat volume or areas at specific body levels, instead of slow and subjective manual ones such as the one that we used, will be required for such large studies.

## 5. Conclusions

We used CT data to measure visceral adiposity of 994 NSCLC patients to obtain a profile of visceral obesity in NSCLC and associate it with disease characteristics. Our work shows that radiological assessment of visceral adiposity is practicable in lung cancer as patients undergo CT routinely, and this allows for the inclusion of visceral adiposity as a meaningful measurement of obesity, unlike BMI, in lung cancer studies.

## Figures and Tables

**Figure 1 cancers-14-03450-f001:**
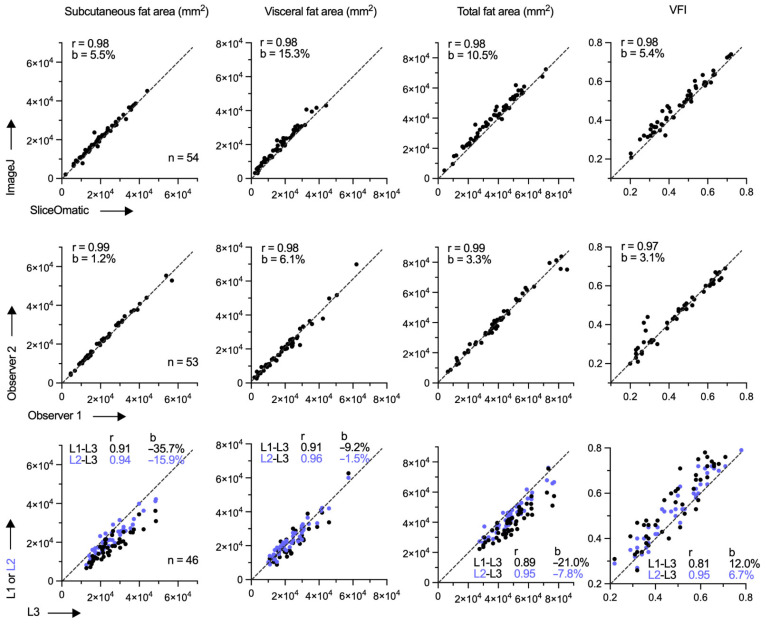
Quantification of abdominal fat in computerized tomography images. Axial cross-sectional body images of non-small cell lung cancer patients at the level of L1, L2, or L3 lumbar vertebrae were analyzed with methods using ImageJ or sliceOmatic software to quantify areas of subcutaneous and visceral abdominal fat deposits. Visceral fat index (VFI) was defined as the ratio of visceral to the sum of subcutaneous and visceral fat tissue areas. Concordance of the measurements between the different software (same observer; n = 54), two observers (using ImageJ; n = 53), and different vertebral levels (same observer using ImageJ; n = 46) is depicted with scatterplots that also show the number of examined cases (n) and values of Pearson correlation coefficient (r) and Bland–Altman bias (b).

**Figure 2 cancers-14-03450-f002:**
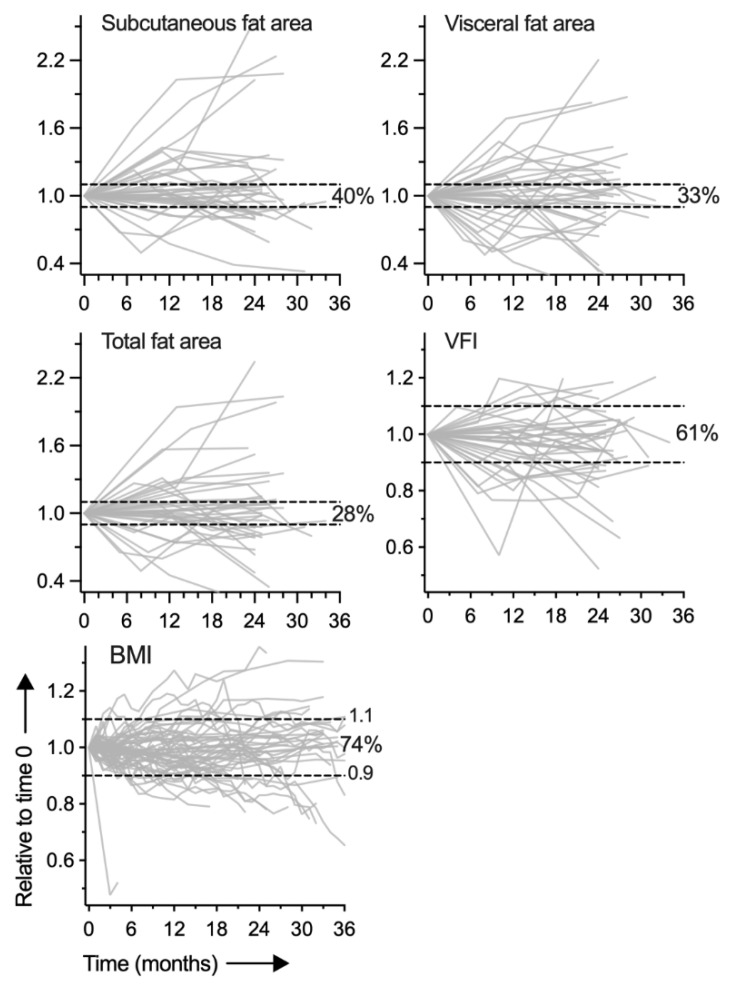
Stability of abdominal fat measures over time. Areas of subcutaneous and visceral abdominal fat deposits at the level of L1 vertebra were measured in axial cross-sectional computerized tomography (CT) images of 37 non-small cell lung cancer patients of the study cohort. For each patient, the CT scans were from three to six different time-points that spanned up to three years. Similar time series data (3 to 36 time-points) were obtained for body mass index (BMI) for a separate subset of the cohort (n = 70). Plots show tracks of the fat measures over time for each patient. The fractions of later time-point values that are within 10% of the value at time 0 are noted. Visceral fat index (VFI) was defined as the ratio of visceral to the sum of subcutaneous and visceral fat tissue areas.

**Figure 3 cancers-14-03450-f003:**
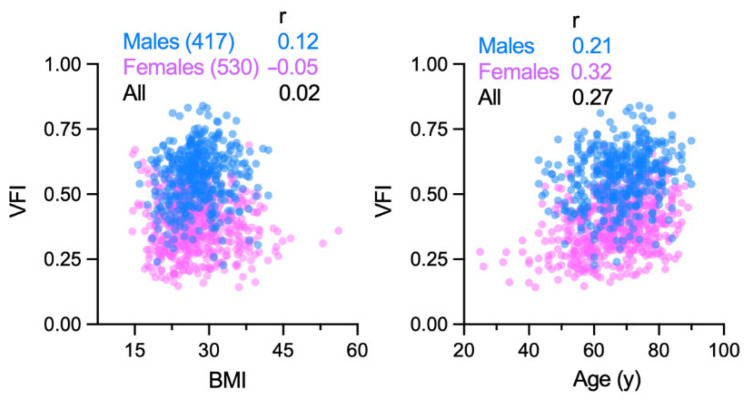
Association of visceral fat index (VFI) with age and body mass index (BMI) in non-small cell lung cancer. VFI was quantified using computerized tomography images of 947 patients at the L3 lumbar vertebral level. The scatterplots illustrate the correlation of VFI with the patients’ age and BMI. Pearson correlation coefficients (r) are shown.

**Figure 4 cancers-14-03450-f004:**
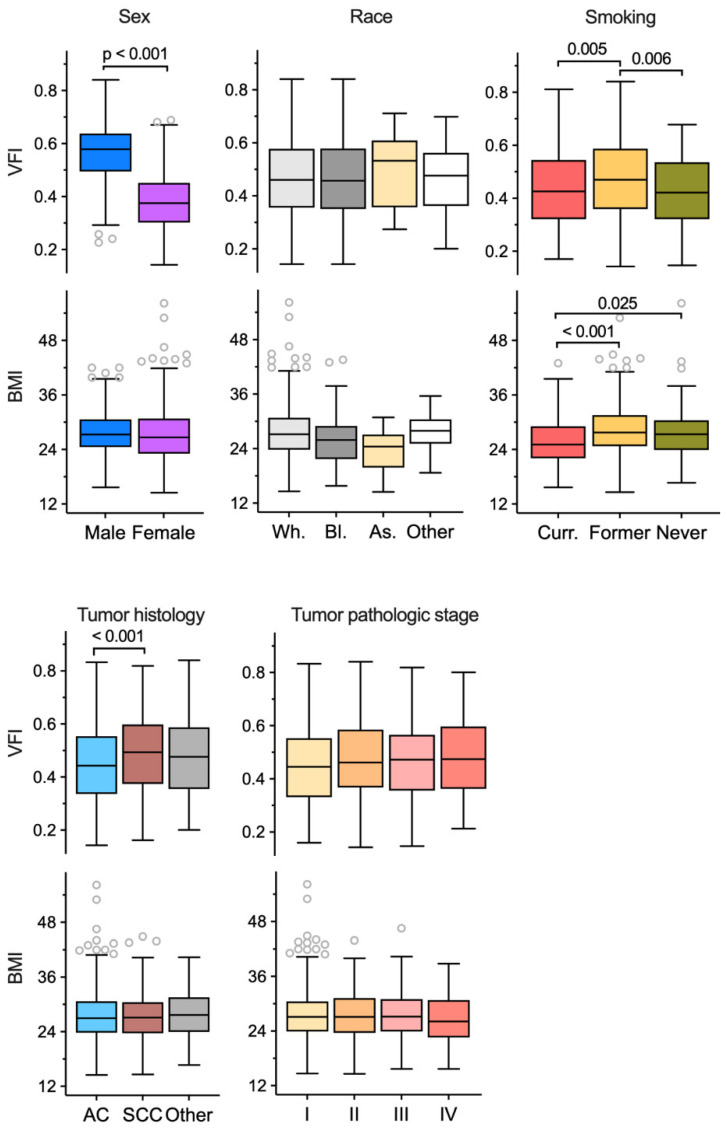
Association of visceral fat index (VFI) with various demographic and clinical features in non-small cell lung cancer patients. VFI was quantified using computerized tomography images of 947 patients at the L3 lumbar vertebral level. Tukey boxplots of VFI and body mass index (BMI) are shown for different subgroups of patients. Median, inter-quartile range, minimum and maximum values of non-outliers, and all outliers are depicted. Welch t and Welch ANOVA (with Games–Howell procedure) tests were used for comparisons, and *p* values ≤ 0.05 in the comparisons are presented. AC, adenocarcinoma; As., Asian; Bl., black; Curr., current; SCC, squamous cell carcinoma; Wh., white.

**Figure 5 cancers-14-03450-f005:**
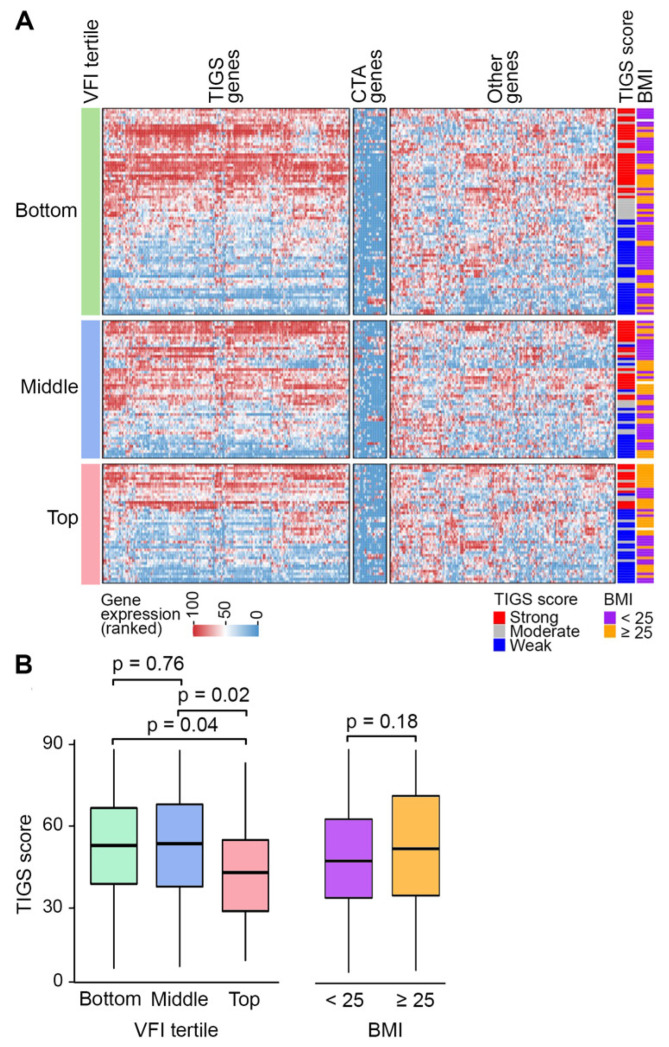
Visceral obesity may attenuate tumor immunogenicity in non-small cell lung cancer. Ranked gene expression measurements of tumors of 175 patients were examined by unsupervised hierarchical clustering to discover three clusters followed by k-means clustering (k = 3). Joint expression of the cluster of tumor immunogenicity signature (TIGS) genes was summarized as TIGS score. (**A**). Tumor gene expression heatmap of the 175 subjects grouped into tertiles using visceral fat index (VFI) as the visceral obesity metric. The three gene clusters include one for 161 TIGS and another for 17 cancer/testis antigen (CTA) genes, and the heat map is annotated with categorized body mass index (BMI) and TIGS score (strong, >61; moderate, 43–61; weak, <44). (**B**). Boxplots of TIGS scores of patients of different VFI and BMI categories. Median, inter-quartile range, and minimum and maximum values of non-outliers are depicted. Wilcoxon rank sum test *p* values are shown.

**Table 1 cancers-14-03450-t001:** Characteristics of the examined population at the L3 vertebral level (n = 947) ^1^.

Characteristic	Median (IQR)/n (%)
Age at time of analyzed CT (years)	66 (59, 74)
Sex	
Female	530 (56.0%)
Male	417 (44.0%)
Race	
White	849 (90.4%)
Black	67 (7.1%)
Asian	11 (1.2%)
Other	12 (1.3%)
Smoking history at time of analyzed CT	
Current	205 (26.0%)
Former	510 (64.8%)
cNever	72 (9.2%)
FEV_1_ (percent-predicted; 276 unknown)	82 (66, 95)
DLCO (percent-predicted; 337 unknown)	75 (62, 89)
Histology of NSCLC tumor	
Adenocarcinoma	585 (61.8%)
Squamous cell carcinoma	251 (26.5%)
Other	111 (11.7%)
Pathological stage of NSCLC tumor	
I	464 (56.3%)
II	200 (24.3%)
III	91 (11.0%)
IV	69 (8.4%)
Body mass index	27.1 (23.7, 30.7)
Body fat area in analyzed CT at L3 level	
Subcutaneous (mm^2^)	19,395 (13,654, 27,628)
Visceral (mm^2^)	17,601 (10,265, 26,112)
Total (mm^2^)	39,679 (27,070, 53,227)
Visceral fat index	0.46 (0.35, 0.58)

^1^ Number and percentage among known values, and median and inter-quartile range (IQR) values are shown for categorical and continuous variables, respectively. CT, computerized tomography; DLCO, diffusion capacity of lungs for carbon monoxide; FEV_1_, forced expiratory volume at 1 s; NSCLC, non-small cell lung carcinoma.

## Data Availability

Fat area measurements of the study cohort along with clinical annotations are provided in Appendix A. Other data may be available from the authors upon request.

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
