# Peer review of "Visceral Obesity in Non-Small Cell Lung Cancer"

_cancers, 2022, doi:10.3390/cancers14143450_

Round 1

Reviewer 1 Report

The authors analyze axial CT image slices to characterize visceral obesity in patients with NSCLC.  They use a large database of 994 patients and then use "arbitrary" subgroups to characterize a variety of features such as optimal axial slice level to use (L3), invariance over observer or imaging software (e.g. ImageJ vs SliceOmatic), changes over time (26% change >10%), correlation between visceral adiposity and other baseline features (sex, race, BMI, smoking status, stage, and histology), and with immunogenicity scores based on VFI tertiles.  

This thorough analysis of visceral fat provides an excellent baseline for future investigations and the authors are to be applauded for their work.  I think the primary utility of this manuscript will be as a reference for additional studies and replication of methods for other researchers.  My main concerns and suggestions are as follows

 - The authors use strong language to emphasize the critical importance of visceral obesity in the study of lung cancer, but nowhere in the paper are outcomes discussed (e.g. survival, progression, cardiac events, etc).  To strengthen the argument they should use language that reflects the contribution of this paper (characterization and robustness of VFI as a potential future covariate) and/or highlight more work in the discussion about the role of visceral obesity in lung cancer patients (e.g. are cardiac outcomes worse, does this have interplay with surgery vs radiotherapy, and, how does this improve prognostication or prediction compared to BMI or weight)

 - An annotated figure in the methods section highlighting the method(s) used to quantify visceral fat could be helpful

 - While it is clearly noted in most places, the authors should highlight in every analysis how big the subgroup was for each component (and which, if any used the entire 994)

 - Did the authors have access to serum lipid testing, which may refine or confound the visceral fat associations?

Reviewer 2 Report

Comments:

  1. How does the author correlate the current findings with their previous one that revealed adverse effects of visceral obesity in patients with NSCLC?
  2. What is the cause of the low correlation between VFI and BMI in NSCLC male patients? Does BMI or smoking play any confounding role? Please discuss.
  3. Also, what is the correlation between VFI and BMI in former smoking NSCLC male patients?
  4. How does the author reconcile their findings with previous research that identified the Body Shape Index (ABSI) as a more precise indicator of abdominal fat and associated lung cancer risk? Please discuss this rationally.
  5. Did the author attempt to measure the visceral fat volume and correlate the findings?
  6. How did the author overcome the variability in the VFI measurement that varies with respiration? Please provide the parameter used for computed tomography imaging of visceral fat.
  7. Can present technique well suited for high-throughput studies of large cohorts for a deeper, more comprehensive coverage? Please discuss this rationally.
  8. What is the current practicability of CT imaging over MRI to assess VFI? Please justify.
  9. The author emphasizes the practicability of measuring visceral obesity by examining routinely collected CT image data. However, radiation exposure for the measurement of VFA is inevitable. Please discuss.
  10. Why did the author focus on a specific anatomical landmark L1-L3 and not L4/L5 to get the most in-depth and accurate results of VFI?
  11. Do authors attempt to define progression-free and overall survival in current study? Please provide evidence.

Round 2

Reviewer 1 Report

Thank you for revising the paper and I have no further recommendations.  I think it is acceptable in this revised form.

Author Response

We thank the reviewer for examining the revised manuscript and finding it suitable for publication.

Reviewer 2 Report

Overall, the authors should be commended on their efforts in addressing the comments, but I am concerned about the low novelty that has been lost due to recent important publications (Barbi et al, Pabla et al,) as pointed out by the author. I think the key message is still difficult to pull out from the manuscript and suggest adding more pathological and demographic characteristics related to distant metastasis and therapy response and correlate it with VFI in lung cancer patients. I think that they should at least attempt to find this information and fill the gaps to ensure the cohort is representative.   

Round 3

Reviewer 2 Report

The authors replied in an adequate way to all my comments, no further suggestions.